# Chemokine CX3CL1 (Fractalkine) Signaling and Diabetic Encephalopathy

**DOI:** 10.3390/ijms25147527

**Published:** 2024-07-09

**Authors:** Mateusz Wątroba, Anna D. Grabowska, Dariusz Szukiewicz

**Affiliations:** Laboratory of the Blood-Brain Barrier, Department of Biophysics, Physiology & Pathophysiology, Medical University of Warsaw, Chałubińskiego 5, 02-400 Warsaw, Poland; mateusz.watroba@wum.edu.pl (M.W.); anna.sepulveda@wum.edu.pl (A.D.G.)

**Keywords:** diabetic encephalopathy, chemokine CX3CL1, fractalkine, central nervous system, neuroinflammation, diabetes mellitus, hyperglycemia, advanced glycation end products, CX3CL1/CX3CR1 axis

## Abstract

Diabetes mellitus (DM) is the most common metabolic disease in humans, and its prevalence is increasing worldwide in parallel with the obesity pandemic. A lack of insulin or insulin resistance, and consequently hyperglycemia, leads to many systemic disorders, among which diabetic encephalopathy (DE) is a long-term complication of the central nervous system (CNS), characterized by cognitive impairment and motor dysfunctions. The role of oxidative stress and neuroinflammation in the pathomechanism of DE has been proven. Fractalkine (CX3CL1) has unique properties as an adhesion molecule and chemoattractant, and by acting on its only receptor, CX3CR1, it regulates the activity of microglia in physiological states and neuroinflammation. Depending on the clinical context, CX3CL1-CX3CR1 signaling may have neuroprotective effects by inhibiting the inflammatory process in microglia or, conversely, maintaining/intensifying inflammation and neurotoxicity. This review discusses the evidence supporting that the CX3CL1-CX3CR1 pair is neuroprotective and other evidence that it is neurotoxic. Therefore, interrupting the vicious cycle within neuron–microglia interactions by promoting neuroprotective effects or inhibiting the neurotoxic effects of the CX3CL1-CX3CR1 signaling axis may be a therapeutic goal in DE by limiting the inflammatory response. However, the optimal approach to prevent DE is simply tight glycemic control, because the elimination of dysglycemic states in the CNS abolishes the fundamental mechanisms that induce this vicious cycle.

## 1. Introductory Overview—Diabetes as a Risk Factor for Encephalopathy

Diabetes mellitus (DM) is a heterogeneous, chronic metabolic disease characterized by elevated blood glucose levels. Hyperglycemia in DM is a consequence of defects in insulin secretion, tissue resistance to insulin or a combination of both these factors [1]. Clinically, two types of diabetes are most commonly diagnosed: type 1 (T1D) and type 2 (T2D). In T1D, the deficiency or lack of insulin is caused by a chronic, progressive autoimmune process that destroys the beta cells of the pancreatic islets of Langerhans, whereas in T2D, tissue insulin resistance occurs, often despite hyperinsulinemia [1]. 

Over time, individuals with glucose metabolism disorders, especially poorly controlled diabetes, are predisposed to serious complications, including cardiovascular disease, kidney disease, blindness, neuropathy, lower-extremity amputation, and diabetic encephalopathy (DE), which is the subject of this review [2,3]. DE is a chronic complication of DM that affects the central nervous system (CNS) and is characterized by cognitive impairment and motor dysfunctions [4]. Therefore, patients with DE are at greater risk of both dementia and postural perturbations (e.g., balance disorders) [5,6]. Moreover, considering the prevalence of diabetes, mainly type 2 diabetes, which is largely influenced by the obesity pandemic, DE has become a common CNS complication of DM, with no effective therapies currently available [7,8,9]. Recent studies have indicated that damage to synaptic mitochondria may play an important role in the pathomechanism of DE, but the nature of this phenomenon remains unclear [10].

### 1.1. Consequences of the Lack of or Insufficient Action of Insulin

The central actions of insulin appear to support cognitive functions by regulating neurotransmitter release, synaptic transmission, and neuronal glucose uptake [11,12,13]. Disturbed brain glucose metabolism and hippocampal insulin resistance may impair cognitive functions and contribute to neurodegeneration [14,15]. In addition, the deficiency of central insulin may reduce cerebral blood flow and blood supply to the cerebral cortex, which can also result in cognitive impairment [16]. Moreover, such deficiency of insulin, resulting either from absolute insulin deficiency or from brain insulin resistance, may promote the accumulation of Aβ aggregates and the hyperphosphorylation of tau proteins because the physiological effects of insulin can oppose these processes [17,18,19,20,21]. Advanced glycation end products (AGEs) can stimulate receptors for advanced glycation end products (RAGEs) and Toll-like receptor 4 (TLR4), which may result in neuroinflammation, the stimulation of nuclear factor kappa-light-chain-enhancer of activated B cells (NF-κB)-dependent signaling pathways, and the release of proinflammatory cytokines [22,23,24,25,26,27]. Therefore, neuroinflammation is a constant component of the pathophysiology of DE [28,29].

### 1.2. The Damaging Effects of Chronic Hyperglycemia

Moreover, chronic hyperglycemia may damage the blood–brain barrier (BBB), mainly through its pro-oxidative effects [24,30]. All these effects cause diabetes, a pathological condition associated with insulin deficiency, to promote neurodegeneration and the occurrence of cognitive deficits, recapitulating to some extent the same pathomechanisms that underlie other neurodegenerative diseases, such as Alzheimer’s disease [31,32,33,34,35,36]. This may justify the introduction of DE, defined as the sum of the effects of insufficient insulin on the CNS. The pathomechanisms of cognitive deficits occurring in DE largely overlap with the pathomechanisms of other neurodegenerative diseases, resulting in neuronal loss [37]. The duration of diabetes increases the incidence of both diabetic neuropathy and DE, which do not immediately appear at the onset of the disease, especially when patients’ blood glucose levels are well stabilized [38]. In patients with diabetes diagnosed less than 12 months prior, the incidence of diabetic neuropathy equals approximately 10%, increasing to 50% 25 years after diagnosis [39]. The most common symptoms of diabetic peripheral neuropathy include paresthesia, numbness, and a burning sensation [40].

### 1.3. The Importance of Chemokine Dysfunction in Diabetes

In humans, the chemokine (or chemotactic cytokine) family includes over 50 small proteins produced by various cells, including those in brain tissue (i.e., astrocytes, microglia, and neurons) [41,42,43]. These secreted proteins interact with metabotropic receptors, also known as G protein-coupled receptors (GPCRs), on the surface of target cells and transmit numerous chemotactic and immunoregulatory signals [44,45]. Since the recruitment of immune cells to the site of inflammation and their subsequent activation and regulation are essential conditions for an effective inflammatory immune response, the chemokine system is a key element in maintaining immune homeostasis [46,47]. Consequently, chemokines are involved in all protective or destructive immune and inflammatory responses, including neuroinflammation [48,49]. In diabetes, changes in chemokine concentrations, activities, and profiles have been demonstrated, mainly involving an increase in pro-inflammatory and neurotoxic effects [50,51,52].

Chemokine CX3CL1 (fractalkine) is commonly found throughout the brain, particularly in neural cells, and its receptor is known to be present on microglial cells [53,54]. As a signaling molecule, CX3CL1 facilitates neuronal glial crosstalk [54,55,56]. 

This review presents the neuroprotective and neurotoxic effects of fractalkine, which may manifest themselves in diabetes and influence the course of DE. Because DE is rarely diagnosed in isolated form (as pure DE), other neurodegenerative diseases that most often co-occur with DE are mentioned.

## 2. Chemokine CX3CL1 (Fractalkine)

### 2.1. Structure

C-X3-C motif chemokine ligand 1 (CX3CL1), also known as fractalkine or neurotactin, is the only member of the δ subfamily of chemokines and seems to bind to only one receptor, CX3CR1, a transmembrane Gi protein-coupled receptor [57]. Many other substances belonging to the chemokine family show less specific binding activity than CX3CL1 [57]. The full-length CX3CL1 molecule is larger than most other chemokines [58] and has two forms. The 95-kDa full-length membrane-bound molecule contains a 76-amino-acid N-terminal chemokine domain, a 241-amino-acid mucin-like glycosylated stalk, a 19-amino-acid hydrophobic transmembrane region (α helix), and a 37-amino-acid intracellular C-terminal domain [42,43]. Another form is a soluble molecule of approximately 70 kDa that contains an N-terminal chemokine domain and an extracellular mucin-like stalk [58,59]. The soluble domain of CX3CL1 acts as a signaling molecule (chemoattractant) and can bind to CX3CR1 receptors expressed on microglia [60]. In contrast, its transmembrane mucin-like stalk may act as an adhesion molecule for microglia and infiltrate leukocytes during the inflammatory response [61,62]. The molecular structures of both forms of FKN are shown in Figure 1.

### 2.2. CX3CL1 in the CNS

In the CNS, CX3CL1 is constitutively expressed in neurons, especially in hippocampal neurons [64], while in astrocytes, its expression can be induced by TNF-α and interferon gamma (IFN-γ) [65]. CX3CR1 receptor activation is associated with several intracellular second messengers. In the brain, CX3CR1 expression is limited to microglia [64]. 

The activation of the CX3CL1-CX3CR1 signaling pathway in microglia, both by the soluble form of CX3CL1 and by its membrane-bound form, inhibits lipopolysaccharide (LPS)-induced major histocompatibility complex class II (MHC2) and cluster of differentiation 40 (CD40) mRNA biosynthesis and the level of interleukin-1 beta (IL-1β) expression, and these anti-inflammatory effects depend on the activation of protein kinase B (Akt) and phosphoinositide 3-kinase (PI3K) [66]. 

The activation of the CX3CL1-CX3CR1 axis also stimulates Akt activation in microglia in a dose- and time-dependent manner. The treatment of primary cocultures of glial cells and neurons with fractalkine results in the transient phosphorylation of Akt within 10 min and extracellular signal-regulated kinase 1/2 (ERK1/2) within 1 min of exposure to CX3CL1 [67]. Moreover, CX3CL1 significantly inhibits neuronal calcium influx induced by N-methyl-D-aspartate (NMDA) receptor activation, and this effect can be abrogated by the inhibition of the ERK1/2-dependent signaling pathway. CX3CL1 also inhibits NMDA-dependent apoptosis through Akt- and ERK1/2-dependent signaling pathways [67]. This effect is likely mediated by the activation of microglia rather than by a direct effect of CX3CL1 on neurons. The treatment of hippocampal neuron cultures with the soluble form of CX3CL1 activates cyclic adenosine monophosphate (cAMP)/Ca^2+^ response element binding protein (CREB), a transcription factor, and ERK1/2 kinase but not kinases such as cJun NH(2)-terminal kinase (JNK) or mitogen-activated protein kinases P38 (P38 MAPK) [68]. 

In addition, it induces the translocation of the NF-κB p65 subunit to the cell nucleus. A specific PI3K inhibitor abrogated the translocation of the NF-κB p65 subunit to the cell nucleus, suggesting that CX3CL1-CX3CR1-dependent signaling activates NF-κB via Akt [69]. These results, however, have not been confirmed in other experimental models and could have been due to the specific cell culture system used, e.g., the contamination of the culture with microglia. 

By acting on the CX3CR1 receptor, CX3CL1 modulates α-amino-3-hydroxy-5-methyl-4-isoxazolepropionic acid (AMPA) receptor phosphorylation, increasing calcium influx and inhibiting excitatory postsynaptic potentials and long-term potentiation (LTP) [70]. 

CX3CL1 may also enhance inhibitory postsynaptic currents, possibly by increasing neuronal responsiveness to γ-aminobutyric acid (GABA) and GABA-dependent chloride anion influx into cells [71]. CX3CL1 can activate the CX3CR1 receptor on microglia with the subsequent release of adenosine, which may, in turn, activate adenosine (A)3 (R) receptors on neurons, inducing a signaling cascade that results in the modulation of GABA-A receptors, increasing their sensitivity to GABA [72]. Adenosine may also activate adenosine A2AR receptors on microglia, inducing the release of D-serine, which acts as a coagonist of the NMDA receptor, increasing calcium influx into cells through NMDA receptor activation [73]. Microglia-derived adenosine may also exert a neuroprotective effect by activating the A1R adenosine receptor in neurons [74]. 

Several studies have shown that CX3CL1 inhibits LPS-induced microglial activation by reducing the production of nitric oxide (NO), interleukin-6 (IL-6), and TNF-α [66,75] and inhibits the neurotoxic effects of LPS-activated microglia in vitro by limiting the release of proinflammatory mediators [75]. These data suggest that high levels of endogenous CX3CL1 expressed in adult CNS neurons lead to the tonic activation of CX3CR1 on microglia and act as a neuronal signal maintaining microglia in a quiescent state [76,77], thus contributing to the neuroprotective role of CX3CL1-CX3CR1-dependent signaling. In contrast, in mixed cultures of neuronal and glial cells collected from CX3CR1^−/−^ mice, as well as in microglial murine BV-2 cells with silenced CX3CR1 production, the LPS-induced release of TNF-α, NO, and superoxide molecules is reduced in comparison to cells from wild-type (WT) mice [78], suggesting that CX3CL1 is also involved in the release of proinflammatory mediators from activated microglia.

## 3. Physiological Role of the CX3CL1-CX3CR1 Signaling Pathway in the CNS

### 3.1. Main CX3CL1-CX3CR1 Signaling Pathways

The attachment of the CX3CL1 molecule to the extracellular determinants of CX3CR1 causes conformational rearrangements preceding the activation of heterotrimeric G proteins (the Gαβγ heterotrimer) of the G protein complex associated with CX3CR1 [79,80,81]. The Gα subunit interacts with G protein regulatory (GPR) domain-containing proteins and synembryn (RIC8), a nonreceptor guanine-nucleotide exchange factor for Gα subunits, to exchange the guanosine-5’-triphosphate (GTP) molecule for guanosine-5’-diphosphate (GDP) [82,83]. The presence of active GTP-bound Gα in the G protein complex leads to its dissociation into Gαi-GTP and a GβGγ dimer. Activated Gαi interacts with downstream effectors [84].

The emerging CX3CL1-CX3CR1 signaling axis utilizes several well-described pathways to activate numerous transcription factors, such as signal transducer and activator of transcription protein (STAT), NF-κβ, and CREB, while inhibiting other factors (e.g., members of the class O forkhead box transcription factor [FOXO]) [85,86,87].

Most CX3CR1-related signaling pathways have been shown to involve other chemokine receptors. These include, among others, the following:Stimulating the mobilization of calcium ions from intracellular resources through the phospholipase C (PLC)/protein kinase C (PKC) pathway [88,89];The activation of appropriate kinases with subsequent downstream signaling within the following pathways:(a)The Janus kinase (JAK)/STAT pathway;(b)The phosphoinositide 3-kinase (PI3K)/protein kinase B (Akt)/IkappaBeta (Iκβ) kinase (IKK)/Iκβ/NF-κβ pathway;(c)Ras kinases (Ras)/Raf kinases (Raf)/mitogen-activated protein kinase kinase (MEK)/extracellular signal-regulated kinase (ERK);(d)MEK kinase (MEKK)/cJun NH(2)-terminal kinase (JNK)/CREB or MEKK/mito-gen-activated protein kinase (P38)/CREB pathways [66,90,91,92].

The above signaling pathways are shown in Figure 2.

The sCX3CL1 molecule is created by cutting off (marked with scissors) extracellular structures (N-terminal chemokine domain and mucin-like stalk) from the membrane form of fractalkine (mCX3CL1) after the action of a disintegrin and metalloproteinase domain-containing protein 10 (ADAM10), tumor necrosis factor alpha (TNF-α) converting enzyme (TACE or ADAM17), matrix metalloproteinase-2 (MMP-2), or cathepsins (CTS). Neuronal sCX3CL1 stimulates transmembrane metabotropic CX3CR1 on microglial cells, causing conformational changes with subsequent G protein activation. During the nucleotide exchange of the guanosine-5’-triphosphate (GDP) for the guanosine-5’-diphosphate (GTP), the activated alpha subunit (Gαi) dissociates from the G protein Gαβγ heterotrimer. Changes in gene transcription after CX3CR1 activation are accompanied by the release of intracellularly stored calcium ions (Ca^2+^) due to the activation of the phospholipase C (PLC)/inositol-1,4,5-triphosphate (IP3)/protein kinase C (PKC) pathway.


*Other abbreviations: Akt—protein kinase B; ERK—extracellular signal-regulated kinase; IκB—inhibitory protein of NF-κB; IKK—IkappaBeta (Iκβ) kinase; JAK—Janus kinase; MEK—mitogen-activated protein kinase kinase; MEKK—MEK kinase; P38—mitogen-activated protein kinases; PI3K—phosphoinositide 3-kinase; Raf—Raf kinases; Ras—Ras kinases.*


### 3.2. Physiological Action of CX3CL1-CX3CR1 Signaling in Brain Tissue

Under physiological conditions, the CX3CL1-CX3CR1 signaling pathway is involved in various brain functions during development and adulthood. Recently, a key role was attributed to synaptic pruning dependent on microglia, which phagocytize inactive synapses during the postnatal maturation of the brain [93]. CX3CR1^GFP/GFP^ mice, in which microglia synthesize green fluorescent protein (GFP) instead of the CX3CR1 receptor, have more synapses than WT mice, at least until the third week of life [93]. In the hippocampal CA1 region, CX3CR1-deficient (CX3CR1^−/−^) mice also exhibit reduced numbers of microglia during postnatal development, suggesting that CX3CL1-dependent signaling may exert a chemotactic effect on microglia in the brain [93]. Therefore, the genetic ablation of the CX3CR1 receptor may cause microglia to become unresponsive to the chemotactic effects of CX3CL1, thereby reducing the number of microglia in the brain. As a result, a greater density of synapses in CX3CR1^−/−^ mice can be explained by a lower number of microglia.

CX3CL1 also inhibits neuronal migration by increasing neuronal binding to the extracellular matrix [94]. However, CX3CL1 has the opposite effect on microglia. Blocking CX3CR1 inhibits microglial cell migration in response to CX3CL1 [95]. This finding supports the hypothesis that CX3CL1-CX3CR1 signaling may act as a pathway guide for microglia, promoting their colonization of the CNS. Moreover, the migration of microglia into the centers of the developing somatosensory cortex, which usually occurs around postnatal day 5, is delayed by several days in CX3CR1^−/−/GFP/GFP^ mice, even if no differences are detected at postnatal day 9 [96]. The absence of CX3CR1 also delays the maturation of functional glutamate receptors [96]. Microglia affect synapse maturation during individual development, which is generally promoted by the CX3CL1-CX3CR1 signaling pathway. CX3CL1 produced by neurons in the adult brain likely maintains microglia in a quiescent, inactivated state. Microglial activation occurs when CX3CL1 release is reduced, e.g., in the hippocampus of aging rats [80]. Such “homeostatic” effects on microglia may play a significant role in the CX3CL1-CX3CR1 signaling pathway.

A strong activation of microglia in response to the intraperitoneal administration of LPS in CX3CR1^−/−/GFP/GFP^ mice has been observed, and apart from that, the transplantation of such activated microglia into WT mice produces completely different effects than the transplantation of microglia collected from CX3CR1^+/−^ mice [82]. Microglia from CX3CR1^+/−^ mice rapidly migrate from the administration site and mainly infiltrate white matter tracts, whereas microglia from CX3CR1^−/−^ mice remain at the site of administration. Moreover, neuronal loss surrounding activated GFP^+^ microglia collected from CX3CR1^−/−^ mice is more significant and more persistent than that in the brains of WT mice that received microglia collected from CX3CR1^+/−^ mice, probably due to increased IL-1β release from microglia collected from CX3CR1^−/−^ mice [97]. In a mouse model of Parkinson’s disease, CX3CR1^−/−^ mice showed more pronounced cell death in the pars compacta of the substantia nigra than did CX3CR1^+/+^ mice, and similar results were obtained for CX3CL1^−/−^ mice. These findings suggest that the CX3CL1-CX3CR1 signaling pathway modulates the activity of microglia and that disruptions in this pathway may result in their impaired function [97].

Citing the results of studies on CX3CL1-CX3CR1 signaling in Parkinson’s disease may provide some approximation to DE, which may be helpful in the absence of studies on animal models regarding such signaling in DE. This is justified by the fact that epidemiological data and clinical trial results suggest that insulin resistance, diabetes, and chronic inflammation contribute to the overlapping etiologies of DE and Parkinson’s disease. Consequently, people with diabetes are around 40% more likely to develop Parkinson’s than those without diabetes [98,99].

In another rat model of Parkinson’s disease [100], CX3CL1 was shown to have a neuroprotective effect and prevented neuronal death in the striatum. Indeed, the administration of CX3CL1 to the striatum of rats is neuroprotective and causes a significant decrease in activated microglia. Similarly, when glial cells and hippocampal neurons were cocultured in vitro, neuronal death was detected when microglia were previously exposed to LPS, and this effect was partially abrogated by the administration of CX3CL1 [101]. The activation of microglia with LPS changes their phenotype from quiescent to phagocytic and neurotoxic.

## 4. The Role of CX3CL1-CX3CR1 Signaling in CNS Pathology

The CX3CL1-CX3CR1-dependent signaling pathway plays a vital role in autoimmune and inflammatory CNS diseases. Multiple sclerosis is a typical autoimmune CNS disease characterized by inflammation and focal demyelination within the spinal cord and brain [102]. An animal model of experimentally induced autoimmune encephalomyelitis represents a disease closely related to multiple sclerosis [103], in which the expression of CX3CL1 and CX3CR1 changes within the sites of demyelination. Indeed, the accumulation of microglia expressing CX3CR1 receptors has been found in brain damage and inflammation in rats with experimentally induced autoimmune encephalomyelitis without any alterations in the neuronal expression of CX3CL1 [104]. However, an increase in CX3CL1 expression has been found in astrocytes located near regions affected by inflammation, which may indicate that astrocytes are the source of excessive CX3CL1 release and attract microglia to these regions [104]. Another aspect refers to the increased expression of CX3CL1 in microglia of rats with experimentally induced encephalomyelitis [73]. In this context, the increase in CX3CL1 expression may be a process by which microglia attempt to return to a quiescent phenotype and inhibit their excessive activation.

Moreover, the disease course of CX3CR1^−/−^ mice with experimentally induced encephalomyelitis is more severe than that of WT mice. These mice also show a more significant expression of proinflammatory cytokines, such as TNF-α and IL-17, than do WT mice [105]. Conversely, concentrations of the anti-inflammatory cytokine IL-10 are significantly greater in WT mice affected by experimentally induced encephalomyelitis than in CX3CR1^−/−^ mice affected by this disease [105]. These results indicate a close correlation between CX3CL1 and CX3CR1 in the regulation of the autoimmune response. An autoimmune response within the CNS may result in the excessive activation of microglia. However, while there is considerable evidence that microglial activation contributes to neuronal damage in multiple sclerosis, there is also evidence that microglia also have essential reparative functions. Microglia can increase the expression of CX3CL1 and CX3CR1, which may constitute a mechanism by which they attempt to prevent hyperactivation and restore the quiescent phenotype in adjacent microglia. Depending on the effectiveness of this autoregulation, microglia may generally acquire a neurotoxic or neuroprotective phenotype. Consistent with this, in the course of multiple sclerosis, one of the polymorphic variants of CX3CR1, namely, CX3CR1^I249/T280^ [106], affects the affinity of CX3CL1 for its receptor and the expression of the receptor itself.

Spinal cord injury significantly damages neurons and completely disrupts axonal continuity, leading to inflammation and neurodegeneration at and around the site of injury [107], which then results in the recruitment of microglia and monocyte-derived macrophages [108]. Microglia and macrophages promote the formation of the glial scar, which reduces the chance of recovering the function of damaged neurons and, thus, the chance of survival of the organism as a whole [109]. CX3CR1^−/−^ mice have a specific subpopulation of macrophages that are not present in WT mice. These macrophages infiltrate the damaged spinal cord and possess unique properties compared to those of macrophages found in WT mice. Microglia in CX3CR1^−/−^ mice produce lower amounts of inducible nitric oxide synthase (iNOS) and IL-6 mRNA after spinal cord injury.

Moreover, in CX3CR1^−/−^ mice, functional recovery after spinal cord injury occurs faster and to a greater extent, suggesting that the relationship between neurons and microglia is in dynamic equilibrium during neuronal regeneration. Therefore, after spinal cord injury, microglia in CX3CR1^+/+^ (WT) mice may release factors that activate astrocytes and promote glial scar formation, inhibiting functional axonal regeneration. The pharmacological blockade of CX3CR1 in an appropriate time window after spinal cord injury may serve as a novel method to inhibit microglial activation and promote neural regeneration. Despite the undoubtedly neuroprotective functions of the CX3CL1-CX3CR1 signaling pathway in the CNS, CX3CL1 may do more harm than good under certain circumstances. Studies conducted in CX3CL1^−/−^ mice to investigate the role of CX3CL1 immediately after ischemic injury suggest that CX3CL1 expression inhibits recovery from ischemic CNS injury [110]. Similar studies in CX3CL1^−/−^ and CX3CR1^−/−^ mice have shown that in both strains of mice, the volume of infarcted tissue after ischemia is lower, and the administration of exogenous CX3CL1 to WT mice reduces the total volume of tissue affected by ischemic infarction. CX3CL1 administration has no effect on CX3CR1^−/−^ mice.

Furthermore, in an in vitro glucose and oxygen deprivation model that reflects in vivo ischemic conditions, CX3CL1 reduced TNF-α release from CX3CR1^−/−^ microglia. These results may explain why administering exogenous CX3CL1 to CX3CR1^−/−^ mice increases the total volume of infarcted tissues, considering the neuroprotective effects of TNF-α [111]. CX3CL1 did not affect TNF-α release in microglia collected from WT mice. Moreover, in CX3CR1^−/−^ mice, the infarct area after ischemia was smaller than that in the WT and heterozygous mice. Greater IL-1β expression has been observed in the astrocytes of CX3CR1^+/−^ mice than in those of CX3CR1^−/−^ mice. This finding suggests that, under stressful conditions, such as during an ischemic episode, CX3CR1^−/−^ microglia acquire an astrocyte function-altering phenotype by default [112].

Studies on the sex-specific effects of CX3CL1-CX3CR1 signaling have shown that, within 12 weeks of an ischemic event in WT and CX3CR1^−/−^ mice, female WT mice recover more functions than female CX3CR1^−/−^ mice [113], while no difference has been found in males. This finding suggests that, for unknown reasons, signaling dependent on the activation of the CX3CR1 receptor has a more significant neuroprotective effect in the event of ischemic episodes in females than in males.

### 4.1. CX3CL1-CX3CR1 Pathway in Aging Microglia

Increasing amounts of data are emerging regarding the role of CX3CL1/CX3CR1 signaling in the aged brain. A significant amount of data suggest that the expression of CX3CL1 in the brain of young rodents is high and decreases with age, which reduces the number of ramified microglia and promotes the release of neuroinflammation markers [114]. Moreover, in old mice exposed to peripheral LPS, the microglial response is enhanced [80,114,115], which may confirm the anti-inflammatory and neuroprotective effects of CX3CL1 in young mice. Interestingly, LPS exposure also reduces CX3CR1 expression in old brains more than in young brains, resulting in a long-term decrease in the expression of these receptors on microglia [116]. Recent studies have confirmed these results and have shown that, while the expression of CX3CR1 on microglia returns to normal within 24 h of exposure to LPS in young mice, this does not occur in old mice [115]. This failure to return regular CX3CR1 expression is accompanied by increased IL-1β release, often exacerbating existing CNS diseases. Taken together, the reduced expression of CX3CL1, CX3CR1, or both proteins in aged brains significantly alters the effectiveness of the signaling axis dependent on these proteins, resulting in both morphological and functional alterations in microglial cell phenotypes, as well as the impairment of microglial function. It is known that neurogenesis in the hippocampus decreases during aging. The pharmacological blockade or genetic ablation of CX3CR1 [86,114] has a similar effect on the dentate gyrus of the mouse hippocampus, with a subsequent IL-1β-dependent decrease in the survival and proliferation rate of neuronal stem cells [114]. In this context, the attenuation of CX3CL1-dependent signaling may contribute to the excessive activation of microglia [114]. It remains to be determined whether the same phenomenon occurs in humans. Considering the reduced neurogenesis in the hippocampus during cognitive impairment and aging, further studies are recommended to determine the involvement of the CX3CL1/CX3CR1 signaling pathway in the pathologies mentioned above in humans. The activation of the CX3CR1 receptor in microglia regulates PI3K activity, reducing IL-1β production [117]. Since aging is characterized by a chronic increase in IL-1β levels in the hippocampus [118] and IL-1β inhibits the cell cycle in neuronal progenitor cells [119], the impaired activation of CX3CR1 receptor-dependent signaling may contribute to the reduced rate of neurogenesis in aged brains, especially because the blockade of IL-1β abrogates these effects [114].

Recent studies have also clarified the role of IL-1β. Sirtuin 1 (SIRT1), a nicotinamide adenine dinucleotide (NAD^+^)-dependent protein deacetylase, has been associated with neuroprotective effects, which are partially dependent on the inactivation of the p65 subunit of NF-κB by SIRT1 and, therefore, on the inhibition of the expression of IL-1β, a protein upregulated by NF-κB [120]. However, the activated CX3CR1 receptor can inhibit the activity of protein kinase A (PKA); thus, the deletion of this receptor may facilitate the activation of PKA and, therefore, the activation of NF-κB, which is also dependent on this kinase [121]. In CX3CR1^−/−^ microglia, SIRT1 activity increases, which likely helps to prevent the excessive activation of NF-κB, but in old brains, it is insufficient to prevent the excessive expression of genes promoted by NF-κB, including the IL-1β-encoding gene [122,123].

Previous studies on various animal models of neurodegeneration have shown that the loss of neuronal interactions with microglia caused by damage to the CX3CL1-CX3CR1 signaling pathway results in a more significant neurotoxic activity of microglia and, therefore, in a more severe course of neurodegenerative diseases [97]. However, it is unknown whether the impaired functioning of the CX3CL1-CX3CR1 signaling pathway occurs as a result or as a cause of the increased activation of microglia, while both scenarios may occur during brain aging or diabetic encephalopathy. No differences in CX3CL1 mRNA expression were detected in hippocampal neurons collected from old rats compared with those collected from young rats, which indicates that posttranslational mechanisms are responsible for the decreased CX3CL1 activity [114]. The administration of exogenous CX3CL1 restores physiological levels of neurogenesis. Moreover, a slight decrease in CX3CL1 expression was observed in middle-aged rats. However, they do not have such an inhibitory effect on the function of the CX3CL1-CX3CR1-dependent signaling axis, as observed in old rats, proving the direct role of aging in the significant inhibition of CX3CL1 expression. In other words, the apparent physiological decline in CX3CL1 expression that occurs during aging may be compensated for during the early but not late stage of this process.

When looking at the interindividual genetic variation in the CX3CR1 coding regions, two single nucleotide polymorphisms (SNPs) can be detected. Interestingly, these polymorphisms are associated with an increased risk of age-related macular degeneration (AMD) [124,125] and a reduced risk of atherosclerosis [126]. Moreover, plasma-soluble CX3CR1 levels are significantly greater in people with mild to moderate Alzheimer’s disease than in people with severe disease. If we assume that the severity of Alzheimer’s disease progresses with age, such observations are consistent with the hypothesis that CX3CL1-CX3CR1 signaling plays a neuroprotective role [127]. Notably, in old mice, voluntary physical exercise increases the CX3CL1 concentration in the brain and, therefore, neurogenesis in the hippocampus [128], which leads to improved hippocampal function [129,130]. Combined, this suggests that decreased physical activity with age may contribute to a decrease in CX3CL1 levels in the brain.

### 4.2. Common Denominators of Brain Aging, Alzheimer’s Disease, and Diabetic Encephalopathy

In the course of both Alzheimer’s disease and untreated diabetes, microglia may be excessively activated by factors such as oxidative stress and neuroinflammation. In Alzheimer’s disease, microglia may be directly activated by extracellular deposits of Aβ aggregates, while in DE, they can be activated by AGEs. Furthermore, damage to the blood–brain barrier occurring in the course of diabetes makes it permeable to substances not generally found in the brain, which may promote neuroinflammation. The pattern of microglial cell activation depends on microglial interactions with neurons, while CX3CL1-CX3CR1 signaling plays a significant role in these interactions. Many studies have indicated that CX3CL1-CX3CR1 signaling may exert a neuroprotective effect by preventing the hyperactivation of microglia and thus the neuroinflammatory response [80,89,90,91]. However, other studies have suggested that CX3CL1-CX3CR1 activation can be harmful in slightly different contexts [110,111,131]. Therefore, the modulation of CX3CL1-CX3CR1 signaling may have different effects depending on the metabolic context [132]. However, much evidence indicates that, in the course of diabetic encephalopathy, the activation of this pathway can exert a neuroprotective effect since it takes part in the inhibition of microglial hyperactivation by neurons, which can prevent neuroinflammation, thus lowering the risk of dementia as a long-term complication of diabetes.

### 4.3. Neuroinflammation and Neurodegeneration in diabetic Encephalopathy

The symptoms of DE consist mainly of cognitive deficits resulting from neuroinflammation and neurodegeneration. One of the possible mechanisms underlying these complications of diabetes is persistent inflammation resulting from the pronounced secretion of proinflammatory mediators and pro-oxidant substances [133]. Proinflammatory mediators are predominantly released from glia, including microglia, astrocytes, and oligodendroglia, in the brain [134,135]. The most common microglia-related function is immune surveillance—both in the healthy brain and in the brain affected by various diseases. Microglia constantly explore their microenvironment by extending and retracting their highly motile processes [136,137]. This property is essential for achieving a rapid response to infections or injuries that lead to the activation of microglia, changing their phenotype from quiescent to activated. At the same time, however, the chronic excessive activation of microglia in the course of diabetes, e.g., due to hyperglycemia, may adversely affect the brain, leading to chronic neuroinflammation. The activation of microglia may occur in response to the disruption of neuronal function, e.g., by excess glycation end products or reactive oxygen species (ROS), and is associated with immunoreactive, morphological, proliferative, and migratory changes in microglial phenotypes [136,138]. The activation of microglia allows the elimination of pathogens and debris from other cells during acute inflammatory reactions, which is a beneficial phenomenon. However, the same activation may have an unfavorable effect on chronic inflammatory reactions, contributing to neurodegeneration [139]. Chronic inflammation within the CNS may result in the excessive activation of microglia, which, under such conditions, may excessively release proinflammatory cytokines and undergo oxidative and nitrosative stress [140]. Activated microglia can proliferate and migrate to sites of brain tissue damage, where they undergo morphological changes and alterations in gene expression resulting from interactions among various signaling pathways [141].

The main pathomechanism of DE comprises dysglycemia-related phenomena, i.e., complications of abnormal plasma glucose concentrations—both too high and too low [142]. Hyperglycemia may result in BBB damage through its pro-oxidative effects, as well as pro-inflammatory actions dependent on AGE-RAGE signaling [143]. Therefore, chronic hyperglycemia—both in the course of type I and type II diabetes—can lead to the hyperactivation of microglial cells, although this effect may be prevented to some extent by CX3CL1–CX3CR1-dependent signaling [80,89,90,91], as already mentioned above.

At the same time, it is noteworthy that hypoglycemia can also be a complication of diabetes, or—to be more precise—of its treatment. Hypoglycemia may occur as a result of insulin overdosage in the treatment of type I diabetes, or as a result of sulphonylurea derivative use in the treatment of type II diabetes. Hypoglycemia may insult the CNS in a completely different way than chronic hyperglycemia, resulting in an acute glucose depletion in neurons and ATP deficiency which may even lead to neuronal death. Thus, hypoglycemia can induce irreversible cognitive dysfunction due to neuronal necrosis, directly independent of inflammatory mediators [144,145,146]. Fortunately, new medications used in the treatment of type II diabetes, such as glucagon-like peptide 1 (GLP-1) analogs, sodium-glucose cotransporter 2 (SGLT-2) inhibitors, and the long-known metformin, are much less likely to induce hypoglycemia, since they do not unconditionally stimulate insulin secretion [147]. On the other hand, hypoglycemia as a complication of type I diabetes treatment can be averted by meticulous measuring plasma glucose concentration and through the precise adjustment of insulin doses.

#### Diabetic Encephalopathy—Focus on Microglia

Much evidence indicates that microglia-dependent inflammation within the CNS plays an important role in the pathogenesis of DE. For example, extracellular nucleotides, particularly adenosine triphosphate (ATP), which act through purinergic metabotropic (e.g., P2Y) and purinergic ionotropic (e.g., P2X) receptors, are critical modulators of microglia–neuron communication [148]. Therefore, microglia may affect the course of DE, among other processes, through interactions with neurons. First, neurons become hyperactive in response to neurotoxic factors, hyperglycemia and hyperlipidemia, after which they release slow-acting microglial activators, such as matrix metalloproteinase-9 (MMP-9), ATP, and chemokines, mainly monocyte chemoattractant protein-1 (MCP-1, also known as chemokine CCL2), and CX3CL1 (fractalkine). Second, the activation of p38 mitogen-activated protein kinases, a class of MAPKs in microglia, produces mediators such as neurotrophins and substances that regulate synaptic transmission and the intensity of inflammation. Microglial inflammation may also result from blocking the interaction between the immunomodulatory molecule CD200 and its receptor CD200R. The CD200/CD200R signaling pathway is responsible for immunosuppressive mechanisms involving the inhibition of macrophages, the induction of regulatory T cells, the switching of cytokine profiles from Th1 to Th2, the inhibition of tumor-specific T-cell immunity, and the induction of myeloid-derived suppressor cells (MDSCs) [149]. The inflammatory response in microglia also occurs due to the activation of signaling pathways related to pattern recognition receptors (PRRs), such as Toll-like receptors (TLRs), a microglial receptor–adaptor complex known as triggering receptor expressed on myeloid cells 2 (TREM2) and DNAX-activating protein of 12 kDa (DAP12), as well as AGE-RAGE signaling [150,151,152,153,154]. Despite having transporters for the three main energy substrates (glucose, fatty acids, and glutamine), during an acute inflammatory response, microglia may experience an energy deficit because microglial energy consumption is dependent on their degree of activity [154]. Because neuronal hyperactivity caused by neurotoxic factors has a feedback-activating effect on microglia, including through the production of CX3CL1, the microglia–neuron interaction in the DE is a vicious cycle (Figure 3).

Therefore, regulating the activity of some signaling pathways within microglia by inhibiting the activation of receptors for ATP (e.g., the purinergic ionotropic receptors P2X4 and P2X7), MMP-9, chemokines (CX3CL1 and CCL2), p38 mitogen-activated protein kinases (a class of mitogen-activated protein kinases (MAPKs)), interleukins (IL-1β, IL-6), and tumor necrosis factor alpha (TNF-α) may contribute to the development of novel treatments for DE [154,155].

## 5. Concluding Remarks

DE is a common long-term and chronic complication of DM. Therefore, with the high and constantly increasing incidence of diabetes, DE contributes significantly to cognitive impairment and motor dysfunctions. A constant component of the DE pathomechanism is neuroinflammation, which is caused by a complete lack of insulin (e.g., in T1D) or the ineffective action of insulin due to insulin resistance (e.g., in type T2D, which most often co-occurs with obesity). In addition to the lack of homeostatic glucose and the anti-inflammatory effects of insulin, which limit NF-κB activation and subsequent proinflammatory cytokine expression, chronic hyperglycemia also contributes to neurodegenerative processes mainly through its pro-oxidative effects, which damage blood–brain barrier integrity and increase neuronal loss. The observed clinical diversity of DE forms can be explained by the fact that aging is the primary factor for most neurodegenerative diseases and that, in many cases, the pathomechanisms of several neurodegenerative diseases overlap (e.g., Alzheimer’s disease and Parkinson’s disease).

Although patients with T2D usually have higher plasma concentrations of CX3CL1 than healthy individuals, due to general pro-inflammatory phenotypes related to a high-carbohydrate hypercaloric diet, there are no research studies that allow verifying if this peripheral CX3CL1 acts on the CNS and thus makes any difference in reference to the course of DE [156]. However, it seems that dysglycemia is the main and fundamental pathomechanism underlying DE.

Fractalkine is an intriguing chemokine with the unique properties of an adhesion molecule (mCX3CL1) and chemoattractant (sCX3CL1), that plays a central role in the nervous system. While neurons constitutively express CX3CL1 in the CNS and it can be induced by TNF-α and IFN-γ in astrocytes, CX3CR1 expression in the brain is limited to microglia. This finding highlights the direction of action of the CX3CL1-CX3CR1 signaling axis, which regulates the level of microglial activity in response to brain injury or inflammation. However, knowledge about the role of CX3CL1 in DE, as well as in other neurodegenerative diseases, remains surprisingly incomplete and controversial. Depending on the clinical context, CX3CL1 may have neuroprotective effects by inhibiting the inflammatory process in microglia or, conversely, maintaining/intensifying inflammation and neurotoxicity. The impact of comorbidities, including CNS aging, should be considered because, as mentioned above, DE does not occur in an isolated form. Therapeutic actions in DE aimed at limiting neuronal hyperactivity, causing impaired synaptic plasticity, should focus on interrupting the vicious cycle within the microglia–neuron interaction involving the CX3CL1–CX3CR1 signaling pathway. This can be achieved both by restoring neural homeostasis and by limiting the inflammatory response of microglia. There is a high probability that influencing the activity of the CX3CL1–CX3CR1 axis may also be beneficial in patients suffering from other diseases predisposing to dysglycemia (e.g., hyperadrenocorticism or chronic autoimmune and inflammatory diseases treated with corticosteroids).

## Figures and Tables

**Figure 1 ijms-25-07527-f001:**
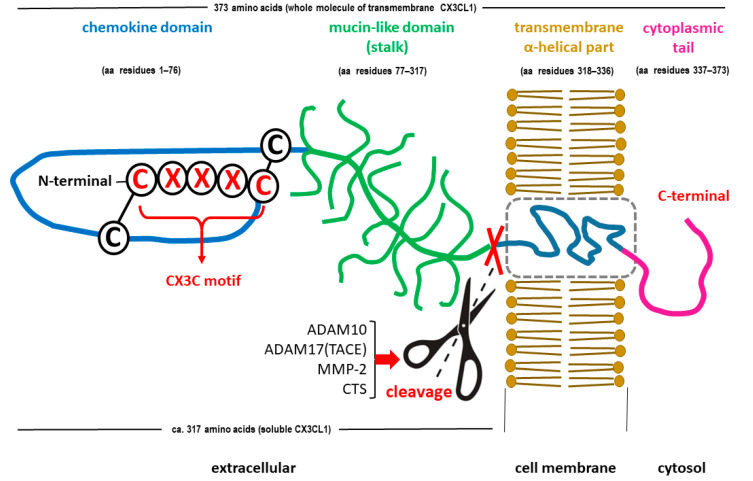
Schematic structure of the C-X3-C motif chemokine ligand 1 (CX3CL1), also known as fractalkine or neurotactin. Both forms of the chemokine are shown: membrane-bound (transmembrane) CX3CL1 and soluble CX3CL1. The soluble CX3CL1, containing an N-terminal chemokine domain and an extracellular mucin-like stalk, is generated through the cleavage of the membrane-bound molecule near the outer surface of the membrane (marked symbolically with scissors and crossed red lines). Release of soluble CX3CL1 may occur upon exposure to a disintegrin and metalloproteinase domain-containing protein 10 (ADAM10), tumor necrosis factor alpha (TNF-α) converting enzyme (TACE or ADAM17), matrix metalloproteinase-2 (MMP-2), or cathepsins (CTS). Adapted from [63].

**Figure 2 ijms-25-07527-f002:**
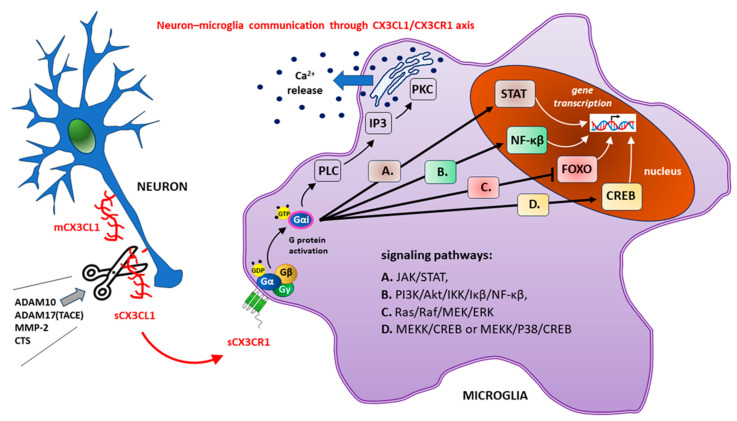
Downstream signaling pathways (A–D) after the activation of the CX3C motif chemokine receptor 1 (CX3CR1) caused by the attachment of the only endogenous ligand, chemokine CX3CL1 (fractalkine), released in a soluble form (sCX3CL1) from neurons. Gene transcription changes are a consequence of the signal transducer and activator of the transcription protein (STAT), nuclear factor kappa-light-chain-enhancer of activated B cells (NF-κB) and cAMP/Ca^2+^ response element binding protein (CREB) activation, while inhibiting the members of the class O of forkhead box transcription factors (FOXO). To keep the figure clear, the interaction of CX3CR1 with other receptors has been omitted.

**Figure 3 ijms-25-07527-f003:**
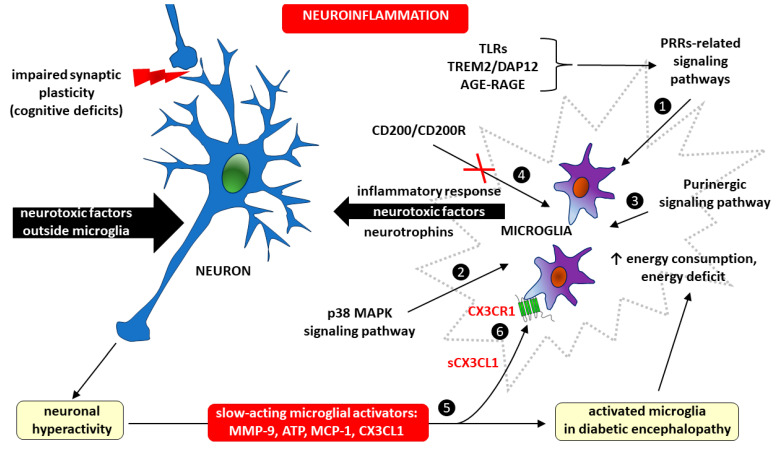
The importance of microglia and CX3CL1/CX3CR1 signaling in the vicious cycle of microglia–neuron interaction during neuroinflammation in diabetic encephalopathy. Microglia activation in diabetic encephalopathy occurs as a result of the action of signaling pathways related to the following: ❶ pattern recognition receptors (PRRs), including Toll-like receptors (TLRs), triggering receptor expressed on myeloid cells 2/DNAX-activating protein of 12 kDa (TRM2/DAP12), and advanced glycation end product/receptor for advanced glycation end product (AGE-RAGE), ❷ the activation of p38 mitogen-activated protein kinases, a class of mitogen-activated protein kinases (MAPKs) p38 MAPK, and ❸ purinergic signaling. Microglial inflammation may also develop as a consequence of blocking the interaction of the immunomodulatory protein CD200 with its receptor CD200R (marked with crossed out red lines) ❹, because the CD200/CD200R signaling provides immunosuppression due to the inhibition of macrophages, the induction of regulatory T cells, the switching of cytokine profiles from T helper-1 (Th1) to T helper-2 (Th2), the inhibition of tumor-specific T cell immunity, and the induction of myeloid-derived suppressor cells (MDSCs) [149]. Neuronal hyperactivity in diabetic encephalopathy may be caused by both neurotoxic factors outside microglia and an inflammatory response within microglia. Neurotoxic agents increase the risk of cognitive deficits due to impaired synaptic plasticity. Whatever the cause, neuronal hyperactivity has a feedback-activating effect on microglia by releasing slow-acting microglial activators, such as matrix metalloproteinase-9 (MMP-9), adenosine triphosphate (ATP), monocyte chemoattractant protein-1 (MCP-1), and fractalkine (CX3CL1) ❺. The soluble form of CX3CL1 (sCX3CL1) has a pro-inflammatory effect by stimulating metabotropic CX3CR1 receptors expressed in microglial cells ❻.

## Data Availability

No new data were created. Instead, the data are quoted from the available cited literature.

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
