# Peer review of "Chemokine CX3CL1 (Fractalkine) Signaling and Diabetic Encephalopathy"

_ijms, 2024, doi:10.3390/ijms25147527_

Round 1

Reviewer 1 Report

Comments and Suggestions for Authors

It is a good review; hopwever, I have a few suggestions, which when employed this can be acepted for publication.

(1) Please distinguish between Type 1 and Type 2 diabetes on how CX3CL1 signalling and diabetic encephalopathy happens in each of the types.  

(2) Please write a paragraph about GL_1-based drugs on how tey afect diabetes and if there are any research related to CX3CL1 signalling when patienst on GLP-1 drugs

(3) Please rewrite the abstract to explain clearly on how you are trying to block the vicious cycle within neuron-microglia interactions as a therapeutic for DE. The writing is not very clear

(4) Are there any other diseases which can benefit from this?

(5) Please discuss if people who can control diabetes through insulin, GLP-1 or lifestyle, also get DE or not. 

Comments on the Quality of English Language

Clarity is required

Reviewer 2 Report

Comments and Suggestions for Authors

Diabetes mellitus increases the risk of micro and macrovascular complications as well as diabetic encephalopathy, defined as a chronic complication characterized by cognitive and motor dysfunction. The reviewed article provides an in-depth overview of diabetes mellitus and its association with diabetic encephalopathy.

CX3CL1, predominantly expressed by neurons, interacts with CX3CR1 on microglia, regulating their activation and neuroinflammatory responses. Chronic hyperglycemia and insulin deficiency contribute to neurodegeneration through mechanisms involving altered glucose metabolism, mitochondrial dysfunction, neuroinflammation, and CX3CL1-CX3CR1 signaling emerges as a critical player in CNS protection, regulating microglial activation processes and neuroinflammatory responses, thus modulating cognitive functions and neuronal survival.

It is known that hypoglycemic episodes in tightly controlled diabetes mellitus can also cause neuronal loss and cognitive dysfunction.  

Please try to insert some information on how does CX3CL1-CX3CR1 signaling influence the neuron survival during repeated episodes of hypoglycemia in diabetes mellitus, and how can this affect cognitive dysfunction.

Reviewer 3 Report

Comments and Suggestions for Authors

The article is well designed and covers the most recent data related to the role of CX3CL1 in diabetic encephalopathy (DE). The topic is very interesting and the summary of the current data on the role of CX3CL1 in the pathogenesis of DE may be useful for further research of this entity. In my opinion, the article could be reorganized for the clarity improvement. The first section should describe major pathophysiological mechanisms of diabetic encephalopathy, then the second section the role of microglia and neuroinflammation, and then the article should be focused on CX3CL1, its physiological roles and roles in DE. Information on the role of CX3CL1 in other pathological states should be very brief to avoid the distraction of the reader.  

Round 2

Reviewer 3 Report

Comments and Suggestions for Authors

I would like to thank the authors for a detailed explanation of their point of view regarding the clarity of the manuscript. I realize that the authors prioritize CX3CL1/CX3CR1 axis, but there is diabetic encephalopathy in the title and the pathogenesis of the diabetic encephalopathy is described in the introductory section. A reader is directed to diabetic encephalopathy, so the article should be focused on the role of CX3CL1 in diabetic encephalopathy. I agree that the authors should describe the structure, the function of CX3CL1, its role in the CNS and diseases and I have no remarks on the organization of chapters. However in chapters 3 and 4 much data regarding Parkinson disease, multiple sclerosis, spinal cord injury are described and this distracts from the main topic and it is CX3CL1 and diabetic encephalopathy. These chapters should be modified and the description of other diseases should be reduced, while more attention should be paid to the role of CX3CL1 in diabetic encephalopathy. Also in the introductory chapter the authors should explain why they decided to review precisely the role of CX3CL1 in diabetic encephalopathy.

Comments on the Quality of English Language

English is very good and the article is written in an understandable manner. 

Round 3

Reviewer 3 Report

Comments and Suggestions for Authors

The authors did their best to meet all my requirements. After this revision I recommend this article for the publication in IJMS.